# The small RNA locus map for *Chlamydomonas reinhardtii*

**Sebastian Y. Müller**[ORCID][☯], **Nicholas E. Matthews**[ORCID][☯¤a], **Adrian A. Valli**[¤b], **David C. Baulcombe***

Department of Plant Sciences, University of Cambridge, Cambridge, United Kingdom

☯ These authors contributed equally to this work.
¤a Current address: The University of Manchester, Manchester, United Kingdom
¤b Current address: Spanish National Centre for Biotechnology (CNB-CSIC), Madrid, Spain
* dcb40@cam.ac.uk

## Abstract

Small (s)RNAs play crucial roles in the regulation of gene expression and genome stability across eukaryotes where they direct epigenetic modifications, post-transcriptional gene silencing, and defense against both endogenous and exogenous viruses. It is known that *Chlamydomonas reinhardtii*, a well-studied unicellular green algae species, possesses sRNA-based mechanisms that are distinct from those of land plants. However, definition of sRNA loci and further systematic classification is not yet available for this or any other algae. Here, using data-driven machine learning approaches including Multiple Correspondence Analysis (MCA) and clustering, we have generated a comprehensively annotated and classified sRNA locus map for C. reinhardtii. This map shows some common characteristics with higher plants and animals, but it also reveals distinct features. These results are consistent with the idea that there was diversification in sRNA mechanisms after the evolutionary divergence of algae from higher plant lineages.

## Introduction

Small (s)RNAs in many organisms are involved in regulation of gene expression at the transcriptional (epigenetic marks) and post-transcriptional (RNA degradation/translational repression) levels, as well as in host defenses against viruses [1]. In eukaryotes, their double stranded or highly structured RNA precursors are processed by Dicer-like (DCL) endonucleases into short 20-25nt RNA duplexes that are bound by Argonaute (AGO) proteins. One of the sRNA strands guides an AGO-containing effector complex toward complementary DNA/RNA to mediate the various mechanisms of RNA silencing [2]. Based on their origin and biogenesis, sRNAs can be classified in diverse classes including small-interfering (si)RNAs, micro (mi)RNA, and piwi-interacting (pi)RNAs [3], but there are many different sRNA subtypes [4] and diffuse boundaries between sRNA classes, as demonstrated for *Arabidopsis thaliana* [5]. Consequently, development and improvement of comprehensive classification methods is required for understanding of sRNA-based regulation networks.

Due to its small genome, vegetative/sexual reproduction, fast growth, motility and capacity to use acetate as carbon source, the photosynthetic green alga *Chlamydomonas reinhardtii*

**Data Availability Statement:** High throughput sequencing data have been deposited in the Array-Express database at EMBL-EBI (www.ebi.ac.uk/arrayexpress) under accession number E-MTAB-8526. All code used for bioinformatic analysis was

deposited on Zenodo (https://doi.org/10.5281/zenodo.3862401).

**Funding:** NEM acknowledges the support of the Engineering and Physical Sciences Research Council (https://epsrc.ukri.org/, grant numbers: EP/M506436/1, EP/M507969/1, and EP/N509565/1). AAV was supported by grant BIO2015-73900-JIN from the Spanish Ministry of Science and Innovation (https://www.ciencia.gob.es/portal/site/MICINN/). SYM was supported by European Research Council Advanced Investigator Grant ERC-2013-AdG 340642 - TRIBE (https://erc.europa.eu/). DCB is the Royal Society Edward Penley Abraham Research Professor (https://royalsociety.org/). The funders had no role in study design, data collection and analysis, decision to publish, or preparation of the manuscript.

**Competing interests:** The authors have declared that no competing interests exist.

(hereafter referred to as Chlamydomonas) has been an important model organism for decades [6] and it was the first unicellular organism in which miRNAs were described [7, 8]. These Chlamydomonas miRNAs, however, are distinct from those of land plants: they have certain animal-like features including their biogenesis and mode of action [9–12]. The Chlamydomonas key proteins in RNA silencing pathways (three AGO and three DCL proteins) are also distinct from homologues in land plants. Phylogenetic, structural and functional analyses indicate a divergence of both protein families since the common ancestor of algae and land plant about 1 billion years ago [10, 12–14]. A striking divergence in the DCLs is the likely loss of PAZ domains in the chlamy lineage [10, 13]. In other organisms and higher plants this PAZ domain is likely to influence the size of DCL cleavage products [15]. Other differences with land plants include (i) the absence in Chlamydomonas of conserved RNA-dependent RNA polymerases (RDRs) that generate dsRNAs from single stranded RNA [16] and (ii) the almost complete absence of non-CG methylation in transposons that is a hallmark of sRNA-directed DNA methylation [17, 18]. Interestingly, the presence of phased siRNAs (phasiRNAs) has been reported in Chlamydomonas despite the absence of conserved RDRs [19], which are important for their biosynthesis in land plants [20].

Taken together, these previously described characteristics of Chlamydomonas sRNA pathways suggest potential divergence from land plants. However, to-date, there has been no comprehensive characterization of the sRNA species found in in this alga. To address this issue we examined Chlamydomonas sRNAs, including miRNAs and siRNAs, based on the distinct loci from which they are produced. We used a Bayesian approach to generate the first comprehensive sRNA locus map for Chlamydomonas. Annotation of the loci based on intrinsic and extrinsic features allowed us to carry out a Multiple Correspondence Analysis (MCA) followed by clustering. We identified 6 classes of sRNAs, which may correspond to distinct RNA silencing pathways. Through comparison of the results with those previously reported for Arabidopsis [5], distinct features to those found in higher plants were uncovered, such as a particular sRNA loci distribution across the genome and their association with the epigenetic landscape. Together, these results help to understand the function of sRNAs in this single-celled alga Chlamydomonas and allow us to hypothesise about the evolution of sRNA-related pathways in green algae and land plants.

## Materials and methods

### Chlamydomonas strains and culture conditions

*C. reinhardtii* strains were obtained from the Chlamydomonas Resource Center (University of Minnesota) and maintained by passing cells into new fresh solid TAP media [2 − *amino* − 2 − (*hydroxymethyl*) − 3 − *propanediol* (TRIS)-acetate-phosphate [21] in the presence of 1,5% agar] every two months, in constant light, at 21 degree celcius.

### Preparation and sequencing of sRNA libraries

Chlamydomonas cells were grown in liquid TAP media with constant shaking at 25 degree celcius under continuous illumination until cultures reached saturation. Total RNA was extracted from cell pellets with TRIzol reagent (ThermoFisher) by following a protocol previously described [7]. sRNA libraries were prepared directly from total RNAs by using the TruSeq v2 RNA Sample Preparation Kit (Illumina) following the manufacturer instructions, and then they were further sequenced on a HiSeq 2000 sequencer. Sequencing data were preprocessed using the ADDAPTS pipeline and tracking system [5, 22]. After 3' adaptor removal, all sequences <15 nt in length were discarded, and the remaining sequences were aligned against the Chlamydonomas genome using the bowtie alignment program tolerating zero mismatches

[23, 24]. Only sequences with at least one perfect match were included in further analyses. The Chlamydomonas reference genome and transcriptome used were Phytozome v5.0 and version 281, respectively.

## sRNA locus map

145 sRNA libraries consisting of 54 replicate groups were used as the basis for analysis. 142 libraries were internally generated laboratory datasets [7, 10] while 3 were from [25]. A locus map for Chlamydomonas sRNAs was produced using the Bioconductor (www.bioconductor. org) package segmentSeq [26]. This package uses a heuristic approach based on sRNA densities to establish an initial locus map which is then refined using Bayesian methods to take into account the separate replicate groups. Sequences that aligned to the genome more than 200 times were excluded from the segmentation. Any gap of greater than 100nt with no reads was sufficient to split a locus. The quality of the segmentation was analysed using a series of diagnostic plots (S1–S4 Figs). The locus map was formed from all loci with a false discovery rate (FDR) of less than 0.05.

## Locus map annotation

The locus map was annotated with intrinsic locus characteristics and publicly available annotations using functions mainly run in the R programming language [27]. The full annotated locus map can be found in S1 File.

## Loci sizes

Chlamydomonas loci were classified into five discrete size classes using class divisions of 100, 400, 1500 and 3000 (corresponding to log 4.6, 6.0, 7.3 and 8.0 respectively) nucleotide width. Class divisions were determined through a combination of the gradient changes seen in the locus size distribution (S5 Fig) balanced with the need to capture biologically relevant size classes. The 100nt cut-off was used to capture loci which might be driven by just one sRNA. The 100-400nt class captures loci which are roughly the same size as an average intron or exon in the Chlamydomonas genome [23]. At 1500nt there is a marked change in the graph and then the 3000nt division captures the long tail of very large loci.

## Predominant 5' nucleotide

To investigate whether there was a predominance for particular 5' nucleotides of the sRNAs originating from each locus, each of the four nucleotides were tested as to whether levels differed significantly from the normal ratio across all loci assuming a binomial distribution [28]. The Benjamini-Hochberg procedure was used to minimise the FDR [29].

## Repetitiveness

Repetitiveness is a measure of the extent to which small RNAs that align to a given location may also align to other genomic locations. We assessed this at each locus using the following equation:

$$R = 1 - \sum_i \frac{x_i}{m_i} / \sum_i x_i \tag{1}$$

where $x_i$ is the number of times the $i$th small RNA within the locus is sequenced and $m_i$ is the number of genomic locations to which that small RNA aligns. This gave a score between 1 and 0 (1 being highly repetitive, 0 not at all repetitive) which was divided into three groups (low

$R < 0.6$, median $0.6 < R < 0.9$, and high with $R > 0.9$) corresponding to the peaks of the distribution shown in S6 Fig.

### sRNA strand bias

Strand bias ratios were calculated from loci in wild type samples with more than five reads. Confidence intervals for strand bias at each locus were calculated assuming a binomial distribution and using a modified Jeffreys interval [28]. Loci were then classified as having a strong ($< 0.2$ or $> 0.8$), medium (between 0.2 and 0.4 or between 0.6 and 0.8) or no bias (between 0.4 and 0.6) (S7 Fig).

### Phasing

The detection of phasiRNA production was undertaken using PhaseTank, a perl based tool which searches for regions of a minimum of four 21nt sRNAs and computes a phasing score for each one [30]. Overlap between sRNA loci and phased regions were then used to annotate loci as phased or not. The phasing score was reported as part of the annotation in S1 File ranging between 0 (no phasing) and 313. This allowed the identification of loci overlapping with a medium phasing region ($0 < $ score $< 60$) or with a high phasing region (score $> 60$) (S8 Fig).

### sRNA size ratios

Since both 20nt and 21nt long sRNAs were shown to be predominant sRNA classes in Chlamydomonas [7] we calculated the ratio between them. Furthermore, there are potential physiological roles proposed for larger [31] and smaller [32] sRNAs in Arabidopsis. We therefore assigned each locus according its predominant sRNA population namely using 20nt and 21nt sizes as thresholds.

### Expression type

Loci were classified as to how ubiquitous their expression was. 10 wild type replicate groups were identified and loci present in more than 5 defined as common (expression:common in Fig 5), between 1 and 5 as inbetween and only 1 as specific.

### Mutant, strain and developmental stage annotation

DCL3 and AGO3 dependence was calculated by determining loci present in at least one wild type replicate group but not in any of the mutant libraries. We also determined loci found specifically in one of the three used strains (CC4350, CC1883, and J3). Presence of loci specifically in libraries of either vegetative or zygotic developmental stages was also calculated.

### Genome annotations

Overlap with various genome features was calculated including predictions for genes, 5' and 3' UTRs, exons and coding sequences obtained from the Phytozome genomics portal (phytozome.jgi.doe.gov) using the most recent Chlamydomonas genome annotation (v5.5) [23]. Promoter regions were calculated as the 500 bp flanking each gene. Transposon locations were established by processing the Chlamydomonas repeat masker file to remove any sequence not explicitly identified as known transposons. Transposons were then classified (S1 Table) according to the unified system proposed by Wicker et al. and using the extensive literature concerning transposon identities [33]. Predictions of miRNAs, inverted repeats (IRs) and tandem repeats (TRs) were sourced from internal lab data with the IRF and TRF algorithms used to identify IRs and TRs respectively [10].

## DNA methylation

Bisulphite sequencing data generated was processed using yama (https://github.com/tjh48/YAMA) with the heuristics based functionality of segmentSeq used to determine loci enriched in CG, CHH or CHG methylation. sRNA Loci were then probed for overlap with these methylation enriched loci.

## Multiple correspondance analysis

MCA was used to cluster the loci according to their annotations using the CRAN (cran.r-project.org) package FactoMineR with the HCPC function adapted to enable K-means clustering [34]. Some annotations were used as supplementary where they were not predictive of the clustering but their correlations were calculated (S2 Table). These supplementary annotations can provide cross-validation of the clustering outputs. The number of dimensions and clusters to select was determined by integrating information from a number of analyses consistent with that applied by Hardcastle et al. [5]. Dimensional reduction techniques like MCA are designed to concentrate the variance explained in the lower dimensions. Thus, at a certain cut-off, higher dimensions can be excluded as not being particularly significant for explaining overall variation. The graphical elbow-method is a common means to do this by considering the % of variation explained for each dimension, with a clear elbow displayed suggesting 6-10 dimensions would be appropriate (S9 Fig). To calculate the stability of the clustering we re-ran the clustering multiple times using random sub-samples derived by random sampling of the loci with replacement to create datasets of the same size as the original. For each iteration of the sub-sampling and clustering we calculated the proportion of sampled loci which retained the same clustering. This enabled us to create box-plots showing the proportion of loci preserving their original clustering across multiple iterations, thus providing an indication of stability. This analysis was repeated for different combinations of dimensions (1-10) and clusters (2-10) (S10 Fig). This analysis showed that clustering was generally stable between seven and ten dimensions for two, three or six clusters. Based on the combination of these two analyses we determined seven dimensions to be appropriate.

To determine whether two, three or six clusters would be most appropriate, we calculated the gap statistic for seven dimensions [35]. Two, three and six clusters all showed large differences between the observed and expected sum-of-squares within the clusters compared to the cluster means (Wss) (S11 Fig). Computing the normalised mutual information (NMI) compared the clustering to annotation feature overlap showed a large increase in NMI for six clusters (S12 and S13 Figs). Thus six clusters was selected for primary analyses.

A cluster hierarchy was also generated by carrying out clustering for two, three and six clusters each time using seven dimensions which allowed the determination of how loci clustered together for lower values of k.

# Results

## sRNAs characteristics in Chlamydomonas

In order to construct a complete locus map we first obtained a comprehensive collection of 145 sRNA libraries encompassing 54 replicate groups (S3 Table). To capture a maximum diversity of sRNA the libraries represent a wide range of conditions, strains and stages of the life cycle. After initial trimming and filtering (see Methods), the libraries contained a total of 22.3 million non-redundant (336 million redundant) sRNA reads mapping to the Chlamydomonas reference assembly genome [23].

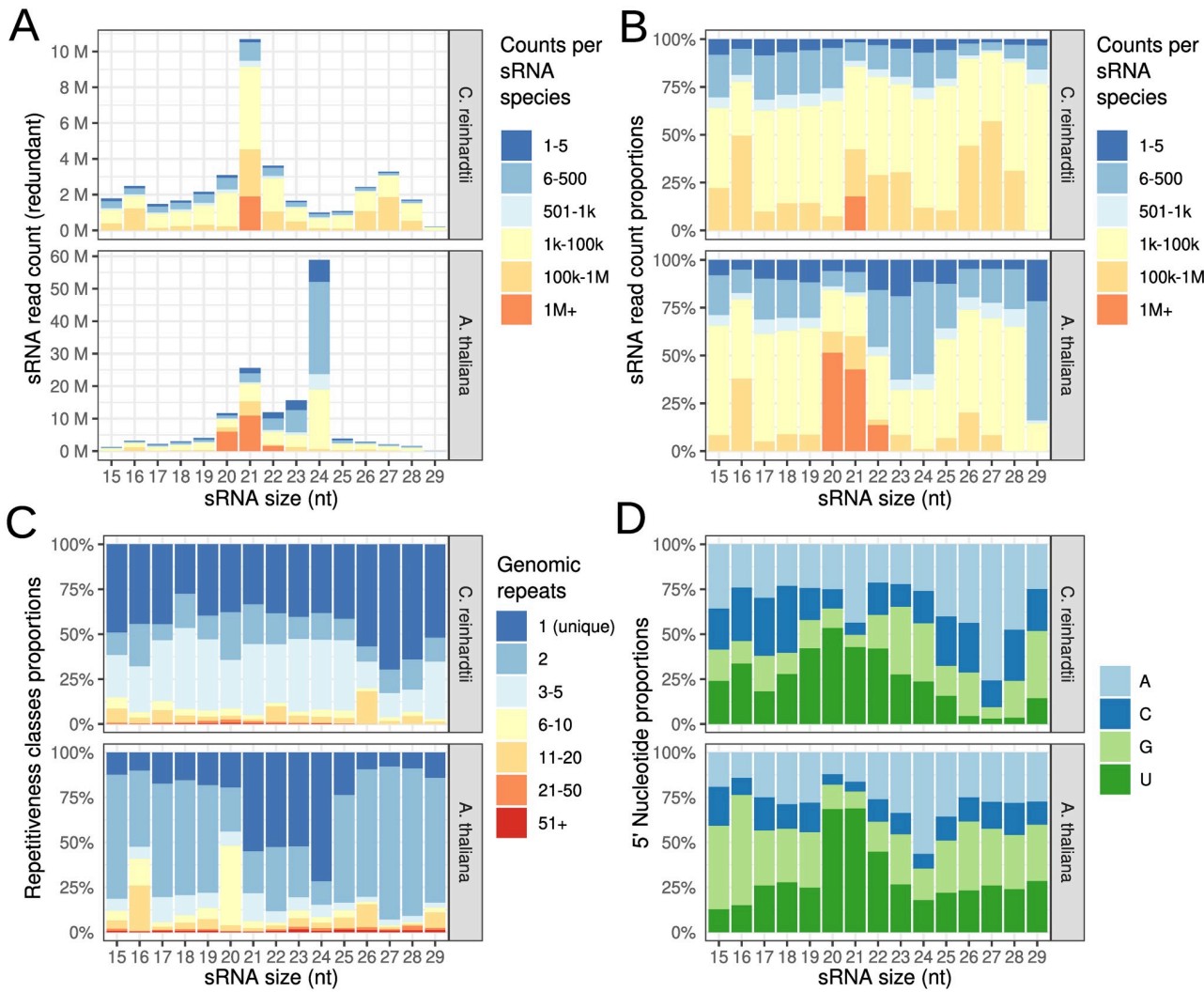

**Fig 1. Characterisation of endogenous sRNA derived from wild-type libraries.** (A) Size distribution of Chlamydomonas and Arabidopsis sRNAs. Redundant read counts were used to quantify the number of sequences obtained for each size class. Color reflects distribution of individual sRNA species abundance. (B) Similar to (A), but fraction of counts instead of total count. (C) Composition of reads mapping to various locations on the genome. Dark blue corresponds to fraction of sRNAs mapping to only one location in the genome, red to 51 or more locations. (D) Sequence composition of the 5' nucleotide of the sRNAs. (sRNAs with multireads greater than 20 were removed).

To obtain a general overview of the sRNA population within our datasets, we first computed the same intrinsic key determinants found to be informative in the sRNA locus classification of Arabidopsis [5]: locus size, 5' nucleotide, repetitiveness of genomic mapping locations and abundance of individual sRNA species. To avoid a bias towards non-representative conditions, for this first part of the analysis we selected only libraries made from wild type strains.

The size distribution of sRNAs (Fig 1A) is dominated by 21nt species comprising 28% of all sRNAs, in line with previous studies [7, 8, 10, 11, 36] and unlike Arabidopsis and other land plants in which the 24nt sRNA fraction predominates (Fig 1A). Also, the 21nt species are the only class exhibiting counts with more than 1 million copies per sRNA species which makes up about 20% within that class (Fig 1B). Many reads (43% redundant and 70% non-redundant) map uniquely to the reference genome, which is comparable to Arabidopsis (52% and 80%),

albeit more evenly distributed across size range as shown in Fig 1C. This difference is most likely due to the absence of a RdDM pathway in Chlamydomonas.

Studies have shown the importance of 5' nucleotide for AGO binding specificity in Chlamydomonas [36] and Arabidopsis [37]. For Chlamydomonas, and in line with previous reports, we found that most sRNAs have adenine (A) and uracil (U) as 5' nucleotide. As in Arabidopsis, however, the 5'-end nucleotide varies greatly for different size classes (Fig 1D). For the predominant 21nt fraction, there was a preference for A and U (26% and 53%) with a higher proportion of 5' A sRNAs in the larger fractions up to 27nt. Overall, we found general agreement of our data with other datasets and we concluded that they are suitable for subsequent locus map generation and classification.

### Defining a comprehensive small RNA locus map

To generate a locus map with these datasets we used the R package SegmentSeq. It employs a heuristic approach based on sRNA densities to derive an initial locus map which is then refined using Bayesian methods to take into account replicate groups [26]. The locus map based on a false discovery rate of less than 0.05 (FDR) had 6164 loci (S1 Fig, S4 Table) and covering 4.1% (4.57Mb) of the reference genome (110Mb). All loci were allocated a unique ID consisting of the CRSL prefix (acronym of Chlamydomonas Reinhardtii SRNA Locus) combined with a successive number starting with CRSL000010.

While the size of the libraries varies markedly, the number of loci per library scales roughly with library size (S2 Fig) and a cumulative analysis indicates that very few extra loci are likely to have been identified with further sequencing (S3 Fig). There was some conservation of loci across replicate groups (S4 Fig), although it should be noted that the libraries consist of a variety of strains, mutants and growth conditions, so complete conservation is not expected. For more stringent FDRs we saw a greater conservation of loci across replicate groups (S14 Fig).

### Small RNA loci annotation

To gain insight into potential function of individual loci, we annotated them based on intrinsic loci features, such as locus size and repetitiveness, and based on features associated with the sRNAs, such as sRNA size, 5'-nucleotide, strand bias and phasing pattern (S2 Table and Methods). In addition, extrinsic annotation features of sRNA loci included sRNA expression in specific conditions, genotype or overlap with genomic features (e.g. genes, transposons, methylation level). In a strong validation of the locus map, all 42 miRNAs previously identified in Chlamydomonas appear as defined sRNA loci [10]. An example is depicted in Fig 2 showing the loci CRSL0041450 which represents miR1157 and miR1157* as part of the 22th intron of the gene Cre12ǧ537671.

Most annotation features are categorical in nature (i.e. overlap with genomic features is either true or false), but others are quantitative (i.e. size and phasing score). In preparation for MCA (see below) the quantitative features were classified into discrete groups according to the modality of the density distributions (S5–S8 Figs). Figure S6 Fig shows a clear bimodal distribution of high or low locus repetitiveness and so we annotated loci in three groups corresponding to the two modes and the intervening section. Strand bias (S7 Fig) shows multiple modal peaks which can be neatly divided into strong bias ($x < 0.2$ and $x > 0.8$) and medium bias (0.2-0.4 and 0.6-0.8). Figure S8 Fig shows phasing to have just a single peak with a cut-off of 60 capturing the modal peak ($< 60$) and the long-tail ($> 60$). Finally, locus size cut-offs were chosen based on marked changes in gradient seen in S5 Fig balanced against a need to capture biologically relevant size classes (see Methods).

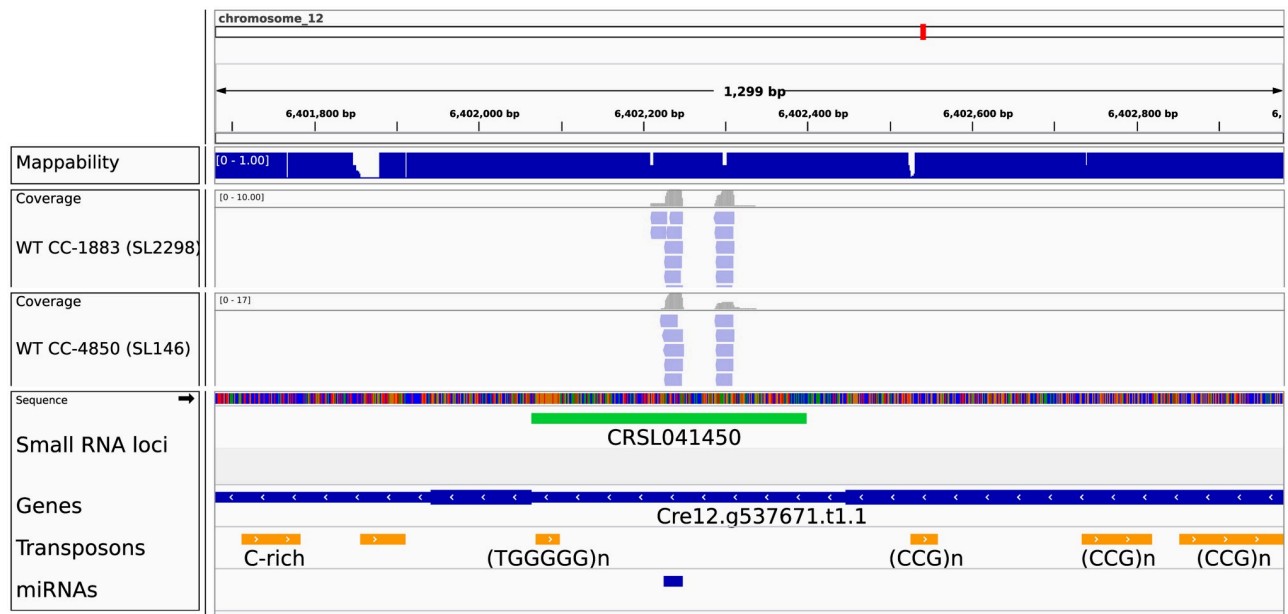

**Fig 2. Genome browser view of a LC3 paragon.** Upper panel shows location of the depicted section within the Chlamydomonas genome. Mappability track is shown below ranging between 0 (low mappability) and 1 (high). Individual mapped small RNAs are shown as red (mapped on + strand) and blue (mapped on—strand) bars for 2 wild type libraries (CC-1883 and CC-4350) along with coverage on top. Genome sequence of the section is shown with a color for each nucleotide (C = blue, G = orange, A = green, T = red). Green bars correspond to individual loci with CRSL prefix followed by a running number. Bottom panel show Genes (blue bars) and Transposons (orange bars).

The overall annotation results are shown in Fig 3 and S5 Table. Most loci have a predominant population of 21nt sRNAs (53%), followed by loci with <20nt sRNAs (22%), 20nt (18%) and <21nt (17%). Notably, 3924 out of 6164 loci (64%) overlap with genes making this feature as the most represented (Fig 3A). These loci overlap (not necessarily exclusively) exons (55%), introns (39%), 3' untranslated region (UTR, 26%) promoters (19%) and 5' UTR (14%). There were also loci overlapping with transposons including both L1 (20%) and other LINE elements (21%), which is consistent with other studies [36]. Most other transposons do not overlap with

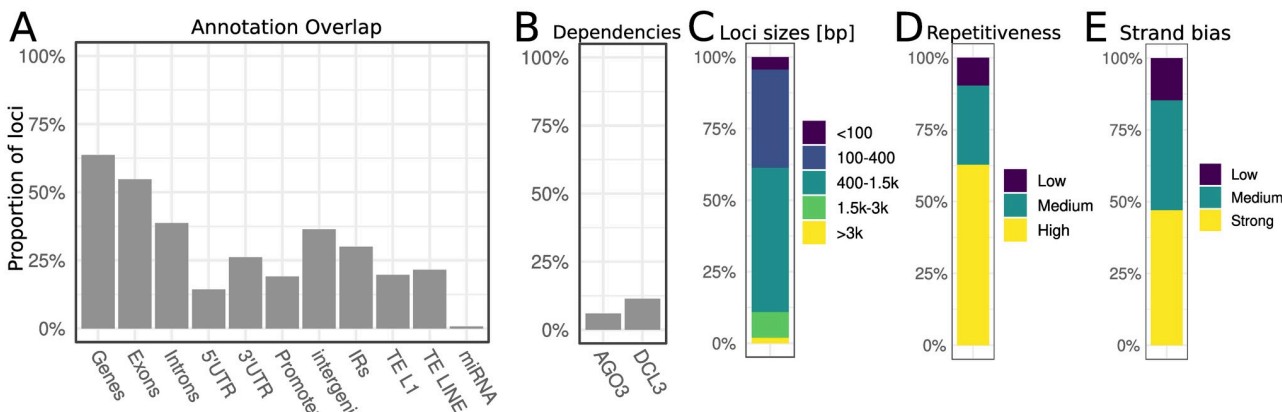

**Fig 3. Proportions of loci characteristics.** (A) Proportion of loci overlapping with various annotations (B) Proportion of loci showing AGO3 or DCL3 dependency. (C-E) show intrinsic locus features. (C) Proportion of loci size classes (D) Proportion of loci repetitiveness classes (E) Proportion of loci stand bias classes. 100% corresponds to all 6164 loci. Abbreviations: UTR (untranslated region), IR (inverted repeats), TE (transposable elements). See Methods for further details.

sRNA loci (S5 Table). Furthermore, only few loci showed evidence of AGO3 dependency (6%,), DCL3 dependency (11%, Fig 3B) and sRNA phasing (2%). The majority of loci are between 400 and 1500nt long (50.4%), followed by 100-400 long loci (34%). Each of the remaining three classes make up less than 10% (Fig 3C). In addition loci were generally found to be very repetitive in their constituent sRNAs with 56% having a high repetitiveness score (Fig 3D). Many loci (41%) show a strong stand bias, where sRNA are found to be predominantly from either the + or—strand, but not both. 35% of loci had a medium bias (Fig 3E).

## Multiple correspondence analysis

Having assigned intrinsic and extrinsic features to each locus (a detailed breakdown of intrinsic and extrinsic features are listed in S2 Table), we used the MCA function from the FactoMineR R package [34] to search for underlying patterns and feature associations. Following dimensional reduction with MCA, k-means clustering was used to group loci according to the annotation patterns (see Methods). We hypothesised that such grouping might identify distinct sRNA types and therefore potentially reveal distinct biogenesis or effector pathways.

To optimise the number of clusters and dimensions to be used we followed an approach used in a similar analysis of Arabidopsis [5] (see Methods). We first evaluated the stability of the clustering using different combinations of clusters and dimensions through random subsampling (S10 Fig). We also calculated the additional variance explained by inclusion of an additional dimension (S9 Fig). Taken together, this analysis indicated that seven dimensions is the optimal number. In addition to the stability tests, computation of the gap statistic [35] and the normalised mutual information (NMI) between the clusters and annotation features both suggested six clusters to be optimal (S11–S13 Figs). The presence of robust clustering for two and three clusters was noted and used to generate a hierarchy plot to demonstrate how loci from clusters grouped together for lower values of k (Fig 4).

The resulting six clusters, referred to as locus class (LC) 1-6, have relatively similar sizes and their association with different annotation features is shown in Fig 5 (also see S15 Fig and S6 Table). All loci with corresponding locus classifications are listed in S4 Table. Associations with annotation features not used predictively in the MCA and K-means clustering (S2 Table) were also calculated and enable some cross-validation of the clustering (S15 Fig). Genome browser views of paragons for each of the six clusters are provided in Fig 2 for LC3 and S16– S20 Figs for LC1, 2, 4, 5 and 6 respectively. The cluster hierarchy (see Fig 4) suggests that the primary division in clusters is between LC1-2 and LC3-6. LC1-2 have a high levels of repetitiveness and a stronger association with genomic methylated regions than LC3-6.

As observed in Arabidopsis [38], the repetitiveness of loci in the genome may correlate with different sRNA silencing mechanisms. LC1 is more associated with transposons (specifically retrotransposons) than LC2. S16 Fig illustrates the overlap of LC1 loci with transposons, low mappability (which corresponds to high repetitiveness) and absence of overlapping genes. Loci were split up mostly due to varying coverage.

LC2 is associated with genic regions, exemplified by a LC2 paragon (CRSL003890) (S17 Fig). Interestingly, a CRTOC1 transposon is superimposed on both the loci and the gene. LC2 contains the largest loci of all classes. Indeed, the shown sRNA locus is 6kb long, which is well outside the normal locus size range.

All miRNA containing loci are in LC3 along with the majority of DCL3- and AGO3-dependent loci. Of the AGO3-dependent loci, 73% have a predominance for U at the 5' end, consistent with AGO3's strong U preference [36]. LC3 loci also demonstrated common expression across wild type libraries and enrichment for 21 nt sRNAs. Interestingly, LC3 contained 130 out of a total of 149 loci that exhibit evidence of phasing.

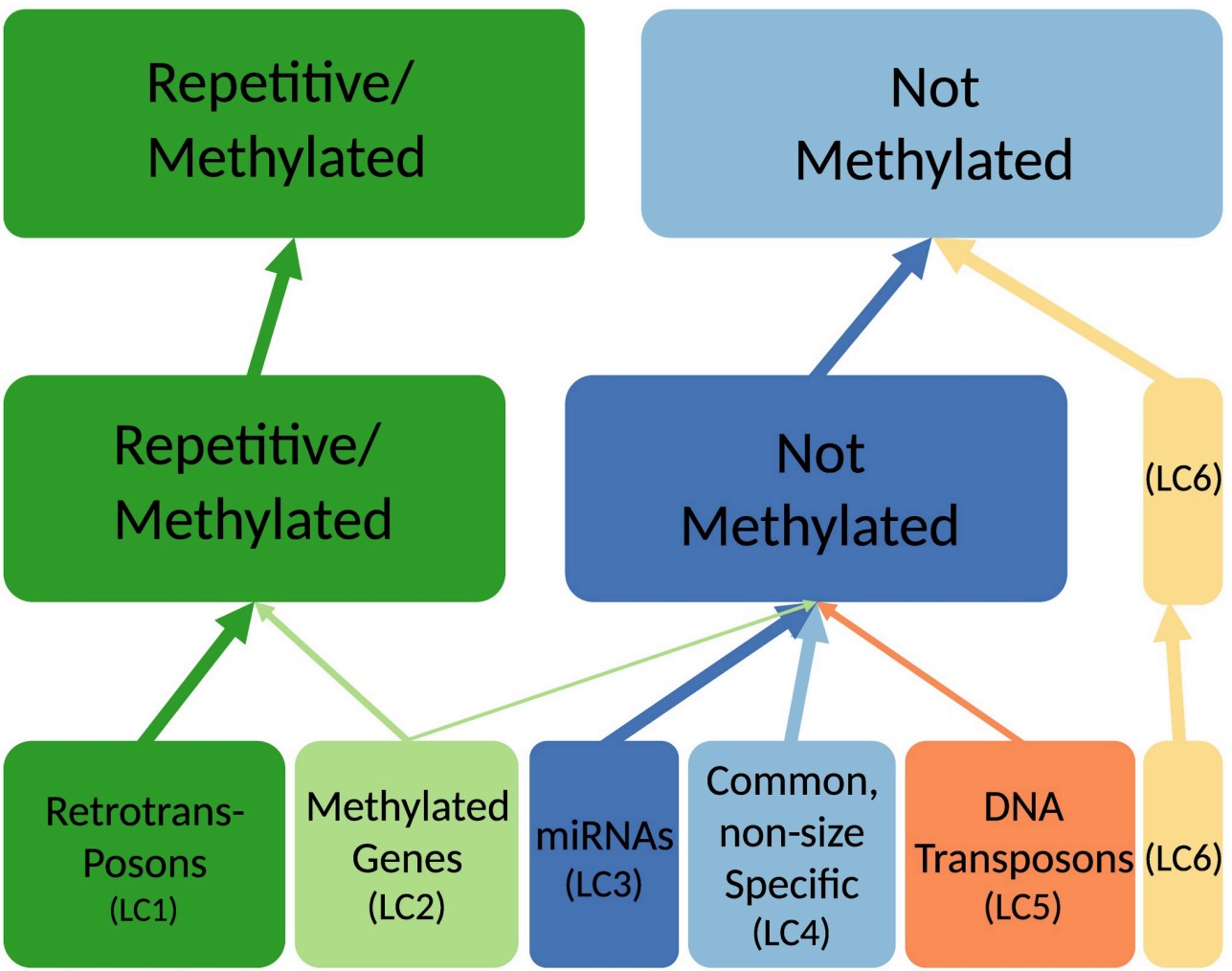

**Fig 4. Cluster hierarchy plot.** Hierarchy of clusters for k = 2,3 and 6. Clusters are annotated with their main distinguishing features. Width of the arrow denotes the proportion of the loci contained within the "higher" cluster.

LC4 sRNAs were similarly represented in most wild type libraries suggesting constitutive or housekeeping roles. However, unlike LC3, there was a lack of sRNA size specificity with 95% having a bias for sRNAs larger or smaller than the modal 20 and 21 nt sRNAs. DNA transposons are associated with 11% of LC5 for which there is a strong bias for sRNAs with a C at the 5'-end potentially indicative of different AGO protein association [39]. There was also a much higher level of LC5 loci specifically expressed during the zygote stage (5%) perhaps indicating roles in silencing DNA transposons at specific points in the life-cycle. LC6 had typically smaller loci (average size 386 nt) as well as most loci (85%) having an enrichment for sRNAs shorter than 20 bp.

Chromosome tracks (Fig 6), demonstrate distinct genomic location patterns for the different LC. LC1-2 are concentrated at the centromere along with higher levels of DNA methylation and a concentration of retrotransposons whereas LC3-6 meanwhile are more evenly spread along the chromosome arms. These patterns are a validation of the locus clusters as chromosomal location was not included as a feature in the MCA.

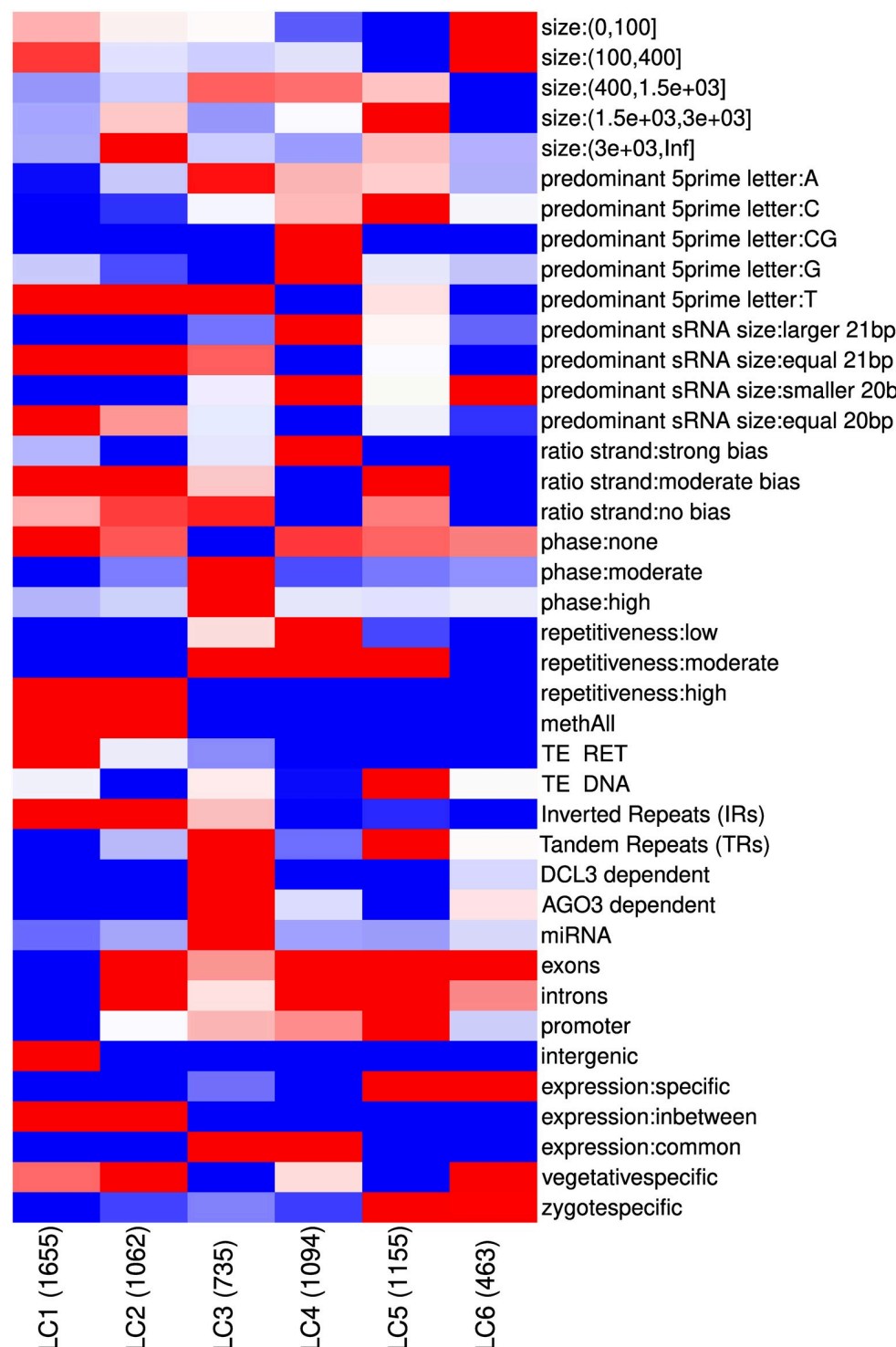

**Fig 5. Heatmap showing association with annotations for the six locus clusters.** Red colours indicate association while blue colours represent disassociation. The size of the clusters is shown in brackets along the x axis.

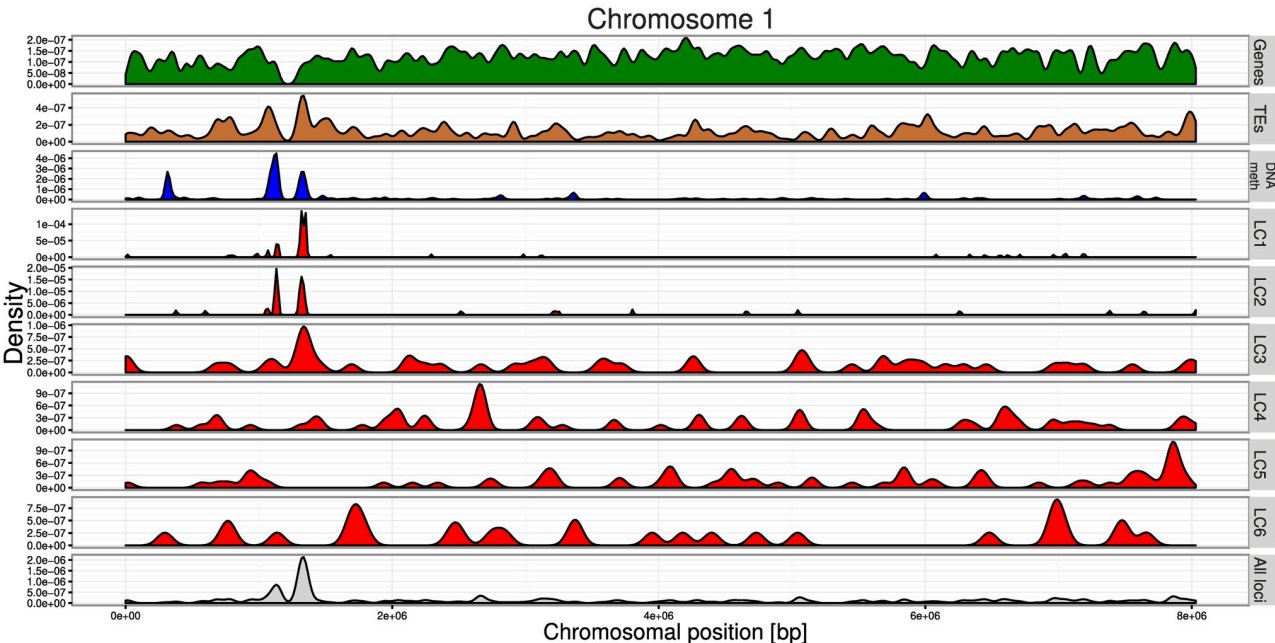

**Fig 6. Chromosome tracks for Chromosome 1.** Density of gene (green), transposon (brown) and methylation (blue) loci are shown on the first three panels. The "DNA meth" track represents combined data for the three methylation contexts from the bisulphite sequencing data. LC1-6 (red tracks) show the locus density of the six clusters while "all loci" (grey) plots the density of all loci.

## Discussion

Chlamydomonas has a silencing machinery more complex than might be expected for a unicellular organism. This complexity precludes a comprehensive characterisation of sRNA-related pathways simply by investigating individual sRNA loci in detail or by studying individual genome-wide features. To address this complexity in this study we followed an approach that allows the data-driven identification of distinct types of sRNA loci [5, 26]. Importantly, the MCA uses a wide range of features as inputs to enable the robust identification of clusters, which could not have been derived by using individual features. Inspection of the feature associations for each cluster enables the validation of the clustering, as well as the dissection of important and unimportant features.

By reporting the first comprehensive map of sRNA loci in a unicellular organism we demonstrate the multi-applicability of our pipeline, which was previously used for Arabidopsis [5, 26], for locus map generation, annotation and clustering. We are mindful that, as with all genome-wide analyses, it is inevitable that there will be some level of noise in the locus map. There is also a possibility that some of the gene-associated loci may correspond to RNA breakdown products rather than bona fide sRNA loci. Yet, our results corroborate (i) the overall size bias for 21nt sRNAs, (ii) the lack of enrichment in the 24nt fraction associated with the RdDM in higher plants, and (iii) the bias for U and A at the 5'-end of sRNAs [7, 8, 10, 36]. Importantly, our analyses groups known Chlamydomonas miRNAs into the same cluster, LC3. Together, these results indicate the robustness of our approach and the overall validity of our findings.

The characteristics of LC3 loci indicate that this cluster includes canonical miRNAs along with other sRNAs, all produced by DCL3-mediated cleavage of precursors and then bound by the AGO3 effector protein. As these non-miRNAs have most, but not all, features

corresponding to *bona fide* miRNAs, we propose that they derive from immature miRNA precursors, which will evolve to either become canonical pre-miRNAs or are on a pathway for potential elimination from the genome. Meanwhile, the computational detection of phasing in some LC3 loci provides further evidence for the possibility of there being phasiRNAs in Chlamydomonas despite its lack of conserved RDRs [19]. If these are in fact true phased loci, this raises the question of whether the Chlamydomonas genome might contain an as yet uncharacterised non-canonical RDR or posses a distinct phasiRNA biogenesis pathway.

LC4 loci are typically independent of DCL3 and AGO3 and, among other peculiarities, they lack bias for U/A at the 5'-end and they are variable in size. The variability in size could indicate imprecise processing by DCL1/2 possibly due to their lack of PAZ domain, which is thought to confer measuring specificity [40]. However, there is also a strong possibility, given the pronounced strand bias and association with genic regions, that the lack of size-specificity of these loci is in fact due to them deriving from RNA breakdown products. In this scenario, a key benefit of the approach taken here is that such loci are clustered together in one locus class, preventing them from undermining the overall results. Further analyses of sRNA species from dcl1/2 and ago1/2 mutants will provide key insight on this matter.

It is likely that LC1-2 are also processed by DCL1/2 because they have a high 21nt bias. In that scenario a PAZ domain-like measuring function may be performed by other dsRNA-binding proteins, as with DGCR8 in the microprocessor complex of animals [41]. An RNA-binding protein DUS16, similarly partners with DCL3 for the proper processing of miRNAs [42, 43].

Our findings also raise questions about DNA methylation and genomic defense from transposons in Chlamydomonas. In Arabidopsis, the RdDM pathway is well characterised [44] and, in the Arabidopsis map, a subset of loci show a combination of RDR2 dependence, bias for 24nt sRNAs (the size class which directs RdDM), and overlap with methylated DNA regions including transposons (the primary targets for RdDM pathway) [5]. The crucial RdDM machinery has not been identified in Chlamydomonas and, in this study consistent with previous findings, no enrichment in the 24nt size fraction was found in any of the sRNA types. However, the presence of loci overlapping with methylated retrotransposons (LC1) and with methylated genic regions (LC2) suggests a possible role of sRNAs during establishment and/or maintenance of methylation states in Chlamydomonas genome. If this connection between sRNA and methylation does exist, then there is a possibility that these loci may represent a distinct form of RdDM in Chlamydomonas that is highly divergent from that of higher plants. Moreover, LC5 loci, with their zygotic-specific expression and DNA transposon overlap, could possibly represent a transposon silencing system activated specifically during the zygotic stage.

Our data-driven approach to identify and classify sRNA loci is intended primarily for the purpose of hypothesis generation, giving possible insights into the biosynthesis and function of sRNAs in Chlamydomonas and, potentially, in other unicellular eukaryotes. The results have allowed the identification of a number of areas for further exploration, as discussed above. Overall, when compared to the previously presented Arabidopsis locus map [5], the results indicate both similarities with higher plants (e.g. LC3) as well as diversification. While these findings relate to just two species, they add to a growing body of evidence suggesting that Chlamydomonas possesses distinct sRNA pathways compared to land plants, with some studies suggesting that they may in fact show more animal-like features [9–12]. Alternatively, these distinct sRNA pathways could have evolved independently. Further studies with mutant strains will enable deeper characterisation aiming to elucidate the functional significance of sRNAs in the unicellular algae Chlamydomonas as well as the evolution of RNA silencing pathways in diverse lineages.

## Supporting information

**S1 Fig. Diagnostic plot 1 for validation of sRNA locus map.** Plot of the number of loci for different FDR levels. Vertical black line corresponds to cutoff used (FDR = 0.05).
(PDF)

**S2 Fig. Diagnostic plot 2 for validation of sRNA locus map.** Number of loci discovered per replicate group plotted against library size.
(PDF)

**S3 Fig. Diagnostic plot 3 for validation of sRNA locus map.** Scatter-plot of number of loci discovered as cumulative sequencing depth increases. Red dots represent WT libraries.
(PDF)

**S4 Fig. Diagnostic plot 4 for validation of sRNA locus map.** Number of loci expressed in a given number of replicate groups.
(PDF)

**S5 Fig. Density plot for log of loci size.** Density plot used to determine cut-offs for locus classification according to locus size. The vertical lines indicate the cut-offs used to classify the loci into discrete classes.
(PDF)

**S6 Fig. Density plot for repetitiveness score.** Density plot used to determine cut-offs for locus classification according to repetitiveness score. The vertical lines indicate the cut-offs used to classify the loci into discrete classes.
(PDF)

**S7 Fig. Density plot for strand bias score.** Density plot used to determine cut-offs for locus classification according to strand bias score. The vertical lines indicate the cut-offs used to classify the loci into discrete classes.
(PDF)

**S8 Fig. Density plot for phasing score.** Density plot used to determine cut-off for locus classification according to phasing score. The vertical line indicates the cut-off used to classify the loci into two discrete classes.
(PDF)

**S9 Fig. Screeplot.** Ranked percentage (y-axis) of variance explained by each dimension (x-axis) of the MCA transformed data.
(PDF)

**S10 Fig. Stability of the cluster results.** Stability of the cluster results (under bootstrapped sampling) achieved for all combinations of dimension selection from 1-8 and all numbers of clusters from 2-10. The y-axis for small plot indicates the proportion of loci which retained their original cluster assignments. The boxplots show the distribution of results after multiple iterations of the bootstrapped sampling and clustering. The x-axis corresponds to cluster result running k-means with (k) clusters.
(PDF)

**S11 Fig. Observed and expected sum-of-squares within each cluster relative to the cluster means (Wss).** The x-axis corresponds to cluster result running k-means with (k) clusters.
(PDF)

**S12 Fig. Non-Mutual Information (NMI) comparing clustering partitioning with transposon superfamily.** The x-axis corresponds to cluster result running k-means with (k) clusters.
(PDF)

**S13 Fig. Non-Mutual Information (NMI) comparing clustering partitioning with annotation feature overlap.** The x-axis corresponds to cluster result running k-means with (k) clusters.
(PDF)

**S14 Fig. Locus conservation and width for loci identified at different FDR thresholds from 0.1 to 1e-4.** The top and middle rows show the frequency (y-axis) of loci found in a given number of replicate groups (x-axis) for the 10 wild type control replicate groups (top row) and all replicate groups (middle row). The bottom row shows locus size density distributions with the log of locus width on the x axis.
(PDF)

**S15 Fig. Heatmap showing associations for all significant annotation features for the six locus clusters.** Red colours indicate association while blue colours represent disassociation. The size of the clusters is shown in brackets along the x axis.
(PDF)

**S16 Fig. Genome browser view of example LC1 paragon loci (CRSL014210-350).** Tracks are annotated as in Fig 2.
(PNG)

**S17 Fig. Genome browser view of an example LC2 paragon loci (CRSL003890).** Tracks are annotated as in Fig 2.
(PNG)

**S18 Fig. Genome browser view of an example LC4 paragon loci (CRSL000070).** Tracks are annotated as in Fig 2.
(PNG)

**S19 Fig. Genome browser view of an example LC5 paragon loci (CRSL009470).** Tracks are annotated as in Fig 2.
(PNG)

**S20 Fig. Genome browser view of an example LC6 paragon loci (CRSL024630).** Tracks are annotated as in Fig 2.
(PNG)

**S1 File. General Feature Format (GFF) file containing all derived loci along with annotations derived in this study.**
(GFF)

**S1 Table. Transposable element classification schema.** Table demonstrates the search-terms used to classify repetitive sequences from the repeatmasker output into transposon superfamilies, orders and classes.
(XLS)

**S2 Table. Overview of annotations used for MCA.** The "type" column indicates whether the annotation is intrinsic (a characteristic of the locus itself) or extrinsic (showing overlap with other genome features of appearing in specific strains/mutants). The final column states whether the annotation primary (i.e. used predictively in the MCA) or supplementary (i.e. not

predictive for the MCA but correlations to clusters calculated).
(XLS)

**S3 Table. Overview of sRNA libraries used to determine the locus map.** Details of replicate groups, genotype, lift cycle stage, and mutant libraries are included.
(XLS)

**S4 Table. List of all sRNA loci including genome position and locus class allocation.**
(XLS)

**S5 Table. Summary of all annotation features showing the number and percentage of loci with a particular annotation.**
(XLS)

**S6 Table. Locus annotations separated by locus cluster.** This table show the number and percentage of loci in each cluster which correspond to a particular annotation. Only annotations with a binary true/false distinction are shown.
(XLS)

**S7 Table. Meta data including ID and library accession for all deposited libraries.**
(XLS)

# Acknowledgments

We would like to thank Attila Molnar, Betty Chung, Daisy Hessenberger and Andrew Bassett for preparing sRNA libraries, and to Thomas Hardcastle for his help during the initiation of this study. We are also grateful to three anonymous reviewers and to Alberto Carbonell for their constructive and insightful comments which helped us to improve and refine this manuscript and would like to thank the Review Commons initiative for facilitating a smooth journal-independent peer-review process.

# Author Contributions

**Conceptualization:** Sebastian Y. Müller, Nicholas E. Matthews, David C. Baulcombe.

**Formal analysis:** Sebastian Y. Müller, Nicholas E. Matthews, Adrian A. Valli, David C. Baulcombe.

**Methodology:** Sebastian Y. Müller, Nicholas E. Matthews.

**Writing – original draft:** Sebastian Y. Müller, Nicholas E. Matthews, Adrian A. Valli.

**Writing – review & editing:** Sebastian Y. Müller, Nicholas E. Matthews, Adrian A. Valli, David C. Baulcombe.

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
