## [Decision Letter · Decision Letter 0]

19 Oct 2020

PONE-D-20-30089

Annotating the algal epigenome: The small RNA locus map for Chlamydomonas reinhardtii

PLOS ONE

Dear Dr. Matthews,

Thank you for submitting your manuscript to PLOS ONE. After careful consideration, we feel that it has merit and it is acceptable pending a minor revision addressing the points raised during the review process.

In particular, the reviewer suggests to remove "epigenome" from the title, but of course you may argue against it in your response if you wish to advocate for small RNAs as part of the epigenome. The reviewer also points out a couple of mistakes that can be easily corrected, and perhaps reducing the number of figures in the main text.

We look forward to receiving your revised manuscript.

Kind regards,

Miguel A Blázquez

Academic Editor

PLOS ONE

Journal Requirements:

2. Please amend either the title on the online submission form (via Edit Submission) or the title in the manuscript so that they are identical.

3.Thank you for stating the following in your Competing Interests section: 

[No].

Reviewers' comments:

Reviewer's Responses to Questions

**Comments to the Author**

1. Is the manuscript technically sound, and do the data support the conclusions?

Reviewer #1: Yes

2. Has the statistical analysis been performed appropriately and rigorously? 

Reviewer #1: Yes

3. Have the authors made all data underlying the findings in their manuscript fully available?

Reviewer #1: Yes

4. Is the manuscript presented in an intelligible fashion and written in standard English?

Reviewer #1: Yes

5. Review Comments to the Author

Reviewer #1: In this work, mostly computational, the authors annotate the sRNA loci in the genome of the unicellular green alga Chlamydomonas reinhardtii. The generated small RNA locus map of C. reinhardtii should facilitate the study of the evolution of small RNA pathways, and their mechanistic diversification compared to those from higher plants. I think the manuscript, although mostly descriptive, is well written and conceived, applies appropriate computational methods (some of them previously published in a previous work of the same lab in Arabidopsis thaliana) and presents a good discussion of the results. The manuscript has already been thoroughly revised in Review Commons, and I have only a few minor comments detailed next.

-I do not think the title should include the word “epigenome”, as the reader may think the paper is about C. reinhardtii epigenome, that is the universe of compounds and proteins that can attach to DNA to turn on or off gene expression, and this is not the main topic of the paper. Therefore, I suggest replacing the title by just the second part of it, which is more realistic: “The small RNA locus map for Chlamydomonas reinhardtii”.

-What is the reason to look for a phasing score if, in theory, no phasiRNAs exist in C. reinhardtii due to the lack of RDRs?

-In Figure 2, labels of the different axes include text of different sizes. Correct and keep labelling consistent across all graphs of the manuscript.

-In Figure 7, there is an orange box with no label. What does it represent?

-The Results section is relatively short in terms of text but contains 9 figures. To have a more balance text/figures ratio I suggest moving at least 1 but preferentially 2 or 3 figures to supplemental. Maybe Figure 7 and/or Figure 8 and/or Figure 9?

6. PLOS authors have the option to publish the peer review history of their article (what does this mean?). If published, this will include your full peer review and any attached files.

Reviewer #1: **Yes: **Alberto Carbonell

---

## [Author Response · Author response to Decision Letter 0]

2 Nov 2020

We broadly agree with the comments of the reviewer (Alberto Carbonell) and greatly appreciate their taking the time to review the manuscript.

Comment #1: “I do not think the title should include the word “epigenome”, as the reader may think the paper is about C. reinhardtii epigenome, that is the universe of compounds and proteins that can attach to DNA to turn on or off gene expression, and this is not the main topic of the paper. Therefore, I suggest replacing the title by just the second part of it, which is more realistic: “The small RNA locus map for Chlamydomonas reinhardtii”.”

• Response: On reflection, we are happy to accept the reviewer’s suggestion. It is important that our title clearly communicates the content of the article and the reviewer makes a valid observation concerning the breadth of what can be considered the “epigenome”.

Comment #2: “What is the reason to look for a phasing score if, in theory, no phasiRNAs exist in C. reinhardtii due to the lack of RDRs?”

• Response: This is an important observation. There are indeed no conserved RDR in the Chlamydomonas genome which might suggest a lack of phasiRNAs in Chlamydomonas. However, there could be a non-canonical RDR. Additionally, phasing has in-fact been previously reported in Chlamydomonas (see this study), although we hadn’t cited this in the present manuscript. As this was an exploratory study, we thought it best to keep an open-mind to the potential presence of some form of phasiRNAs and corresponding phasing. We recognise that we could have made these arguments explicitly in the manuscript and have therefore added text into the introduction to highlight the phasiRNAs have been previously identified (lines 31-34) and a brief discussion of the possible implications of this (line 363-369).

Comment #3: “In Figure 2, labels of the different axes include text of different sizes. Correct and keep labelling consistent across all graphs of the manuscript.”

• Response: Thank you for pointing this out, we have now tidied up the figure labels on this figure.

Comment #4: “In Figure 7, there is an orange box with no label. What does it represent?”

• Response: Good spot. We’ve now added a label to the orange box.

Comment #5: “The Results section is relatively short in terms of text but contains 9 figures. To have a more balance text/figures ratio I suggest moving at least 1 but preferentially 2 or 3 figures to supplemental. Maybe Figure 7 and/or Figure 8 and/or Figure 9?”

• Response: Thanks for this helpful piece of feedback. We agree that moving some figures to the supplemental would tidy up the flow. As for which, we thought it best to move the plots which were more focussed on validation or “diagnostics” rather than core results. We therefore decided to move figures 2, 4 and 6 to the supplementary.

---

## [Editor Report · Decision Letter 1]

4 Nov 2020

The small RNA locus map for *Chlamydomonas reinhardtii*

PONE-D-20-30089R1

Dear Dr. Matthews,

We’re pleased to inform you that your manuscript has been judged scientifically suitable for publication and will be formally accepted for publication once it meets all outstanding technical requirements.

Kind regards,

Miguel A Blázquez

Academic Editor

PLOS ONE
---

## [Editor Report · Acceptance letter]

11 Nov 2020

PONE-D-20-30089R1 

The small RNA locus map for *Chlamydomonas reinhardtii*

Dear Dr. Matthews:

I'm pleased to inform you that your manuscript has been deemed suitable for publication in PLOS ONE. Congratulations! Your manuscript is now with our production department. 

Kind regards, 

on behalf of

Dr. Miguel A Blázquez 

Academic Editor

PLOS ONE